# Inflammation Regulation via an Agonist and Antagonists of α7 Nicotinic Acetylcholine Receptors in RAW264.7 Macrophages

**DOI:** 10.3390/md20030200

**Published:** 2022-03-09

**Authors:** Yao Tan, Zhaoli Chu, Hongyu Shan, Dongting Zhangsun, Xiaopeng Zhu, Sulan Luo

**Affiliations:** 1Medical School, Guangxi University, Nanning 530004, China; tanyao971006@163.com (Y.T.); dlhy34@163.com (H.S.); 2Key Laboratory of Tropical Biological Resources of Ministry of Education, School of Pharmaceutical Sciences, Hainan University, Haikou 570228, China; 20100700210003@hainanu.edu.cn (Z.C.); zhangsundt@163.com (D.Z.)

**Keywords:** α7 nAChR, agonist PNU282987, antagonists α-conotoxin [A10L]PnIA and MLA, inflammation, IL-6

## Abstract

The α7 nicotinic acetylcholine receptor (nAChR) is widely distributed in the central and peripheral nervous systems and is closely related to a variety of nervous system diseases and inflammatory responses. The α7 nAChR subtype plays a vital role in the cholinergic anti-inflammatory pathway. In vivo, ACh released from nerve endings stimulates α7 nAChR on macrophages to regulate the NF-κB and JAK2/STAT3 signaling pathways, thereby inhibiting the production and release of downstream proinflammatory cytokines and chemokines. Despite a considerable level of recent research on α7 nAChR-mediated immune responses, much is still unknown. In this study, we used an agonist (PNU282987) and antagonists (MLA and α-conotoxin [A10L]PnIA) of α7 nAChR as pharmacological tools to identify the molecular mechanism of the α7 nAChR-mediated cholinergic anti-inflammatory pathway in RAW264.7 mouse macrophages. The results of quantitative PCR, ELISAs, and transcriptome analysis were combined to clarify the function of α7 nAChR regulation in the inflammatory response. Our findings indicate that the agonist PNU282987 significantly reduced the expression of the IL-6 gene and protein in inflammatory macrophages to attenuate the inflammatory response, but the antagonists MLA and α-conotoxin [A10L]PnIA had the opposite effects. Neither the agonist nor antagonists of α7 nAChR changed the expression level of the α7 nAChR subunit gene; they only regulated receptor function. This study provides a reference and scientific basis for the discovery of novel α7 nAChR agonists and their anti-inflammatory applications in the future.

## 1. Introduction

Nicotinic acetylcholine receptors (nAChRs) are ligand-gated ion channel proteins regulated by acetylcholine (ACh) and are widely distributed in muscles and the central and peripheral nervous systems. nAChRs consist of five transmembrane subunits, and 12 neuronal nAChR subunits have been cloned, including nine α subunits (α2–α10) and three β subunits (β2–β4) [1,2,3]. α7 nAChR, an important homopentamer, is widely distributed in the mammalian central nervous system (CNS), including the cerebral cortex, olfactory bulb, and hippocampus. It is closely related to neurological diseases such as Alzheimer’s disease (AD), epilepsy, schizophrenia, Parkinson’s disease (PD), and brain inflammation [4,5,6,7]. In addition, α7 nAChR is also expressed on peripheral cells such as macrophages, lung cancer cells, cardiomyocytes, and vascular endothelial cells and is a key protein in the cholinergic anti-inflammatory pathway [8,9,10,11,12]. Studies have shown that α7 nAChR activation can prevent the cleavage of inhibitor of nuclear factor kappa-B (IκB) and the nuclear factor kappa-B p65 (NF-κB p65) in the nuclear factor kappa-B (NF-κB) signaling pathway. At the same time, it can catalyze the cleavage and phosphorylation of janus kinase 2 (JAK2) and activate the JAK2/signal transducer and activator of transcription 3 (JAK2/STAT3) pathway. Finally, phosphorylated STAT3 binds to the p65 subunit of NF-κB and blocks the transcription of NF-κB, which inhibits inflammatory cytokine (interleukin-1β (IL-1β), interleukin-18 (IL-18), tumor necrosis factor α (TNF-α), etc.) and chemokine (platelet-derived growth factor (PDGF), interleukin-18 (IL-8), monocyte chemoattractant protein-1 (MCP-1), fibronectin (FN), etc.) production and release [13,14]. Previous studies have shown that the cholinergic anti-inflammatory pathway regulated by α7 nAChR is a key pathway for the treatment of pneumonia and that α7 nAChR can affect the proliferation, angiogenesis, metastasis, and apoptosis of lung cancer cells, gastric cancer cells, colon cancer cells, and other malignant tumor cells [15,16,17]. An in-depth study on the role and mechanism of α7 nAChR in the cholinergic anti-inflammatory pathway is an important basis for the design of α7 nAChR-related anti-inflammatory drugs.

Inflammation is a common pathological process that also protects the body’s immune system by eliminating harmful stimuli or pathogens and promoting tissue repair. The traditional view suggests that the inflammatory response is regulated by humoral pathway-inflammatory factors, and anti-inflammatory factors produced by immune cells together maintain the balance of proinflammatory and anti-inflammatory responses in the body [18]. However, when the balance is disrupted, inflammation is uncontrolled, possibly leading to immunosuppression. In recent years, studies have found that, in addition to bodily fluids, the nervous system is involved in the regulation of the inflammatory response; specifically, the abovementioned α7 nAChR-regulated cholinergic anti-inflammatory pathway is involved [19]. Whether in the humoral inflammation regulation pathway or the cholinergic anti-inflammatory pathway, macrophages are important effector molecules that engulf foreign bodies, present antigens, and secrete cytokines. Macrophages are differentiated from monocytes that have been differentiated from myeloid precursor cells in peripheral tissues [20,21]. Activated macrophages can be broadly classified into classically activated macrophages (M1) and alternately activated macrophages (M2) [22,23,24]. M1-type proinflammatory macrophages can cause excessive production of proinflammatory cytokines, leading to immune imbalance and promoting the development of inflammation in the body, while type M2 macrophages have been proven to relieve inflammation and promote tissue damage repair. Inflammation is involved in the development of many diseases, such as atherosclerosis, autoimmune hepatitis, asthma, and pneumonia [25,26,27]. M1 macrophages play important roles in the pathological process of some diseases, such as diabetes, atherosclerosis, and tumors [27,28,29]. Recently, Wang Z et al. (2020) found that inulin can promote the polarization of M1 macrophages to the M2 phenotype, thereby attenuating proinflammatory factors such as IL-6 secreted by M1 macrophages and promoting the secretion of the anti-inflammatory factor IL-10, which can reduce inflammatory damage in alcoholic liver disease [30]. Therefore, understanding and intervening in the inflammatory regulatory process of M1 macrophages is of great significance for the treatment of many inflammation-related diseases.

In this study, lipopolysaccharide (LPS) was used to induce RAW264.7 mononuclear macrophages to differentiate into M1-type macrophages, which comprised the inflammatory cell model used for experiments. The expression pattern of various nAChR subunits on normal and inflammatory macrophages was identified. Furthermore, an α7 nAChR agonist (PNU282987) and antagonists (MLA and α-conotoxin [A10L]PnIA) were applied to explore the regulation of α7 nAChR on M1-type macrophage inflammation.

## 2. Results

### 2.1. Identification of nAChR Subunit Expression in RAW264.7 Cells

To identify the expression type and level of the nAChR subunits in RAW264.7 cells, quantitative PCR (qPCR) analysis was performed on four biologically duplicated normal RAW264.7 cells (Appendix A). The expression of ten subunits (α3, α4, α5, α6, α7, α9, α10, β2, β3, and β4) was identified in RAW264.7 cells (Figure 1). Among these subunits, α3, α4, α7, α10, and β2 had higher expression levels (Ct ≤ 25), and α5, α6, α9, β3, and β4 (Ct > 25) had lower expression levels (Figure 1a,b). The analysis of the expression (relative decrease in fold change) of other subunits relative to the α7 subunit showed that there was no significant difference between the expression of the α7 subunit and most of the other subunits, except for α6 and α9, which was downregulated (relative decrease in fold change was 16.8 and 7.0, respectively). The results are shown in Table 1 and Figure 1c.

### 2.2. Differential Gene Expression Analysis of Various nAChR Subunits in Inflammatory RAW264.7 Cells

To identify the expression of various nAChR subunits in inflammatory RAW264.7 cells, we used an LPS-induced inflammation model (Appendix A) and used qPCR to evaluate the expression level of nAChR subunit genes in RAW264.7 macrophages after LPS induction (Figure 2). Similar to normal RAW264.7 cells, inflammatory RAW264.7 cells expressed ten nAChR subunit types (α3, α4, α5, α6, α7, α9, α10, β2, β3, and β4) (Figure 2a). Among these subunits, α10 and β2 were expressed at higher levels (Ct ≤ 25), and the expression level of the α7 nAChR subunit gene was moderate (Figure 2a,b). The analysis of the decrease in fold change of other subunits relative to the α7 subunit showed that only the expression of the α6 subunit gene was significantly different from that of the α7 subunit (relative decrease in 25.7-fold change), and the results are shown in Table 1 and Figure 2c. Although most of the subunits (except β3) tended to be expressed at low levels in inflammatory macrophages compared with normal cells (Figure 2d), the expression of the ten nAChR subunit genes was not significantly reduced in the analysis of significant differences (Figure 2e). This outcome means that the levels of nAChR subunits expression in inflammatory and normal cells were not significantly different.

### 2.3. Synthesis of α7 nAChR Antagonist α-CTx [A10L]PnIA

To study the regulatory function of α7 nAChR in macrophages on inflammation, we selected both the α7 agonist PNU282987 and the specific antagonists MLA and α-CTx [A10L]PnIA. The synthetic [A10L]PnIA linear crude peptide was purified and oxidatively folded to obtain the active peptide (Figure 3). The [A10L]PnIA linear crude peptide (purity of ~80%) was purified by HPLC, yielding purity greater than 95% (Figure 3b, Peak 1). The active peptide of [A10L]PnIA was obtained by two-step oxidative folding in vitro (Figure 3a) with an elution time of 11.48 min (Figure 3b, Peak 2). Compared with that of the linear peptides, the elution time of [A10L]PnIA showed no obvious change before or after oxidative folding. The hydrophobicity of [A10L]PnIA did not change significantly before or after the oxidative folding procedure. The molecular weight of [A10L]PnIA was identified by ESI-MS. The molecular weights of the linear and oxidized peptides were 1811.32 Da and 1665.20 Da, respectively (Figure 3c,d); the observed monoisotopic masses were consistent with the theoretical calculation.

### 2.4. CCK-8 Assay to Identify Cytotoxicity

The α7 nAChR agonist PNU282987 and antagonists MLA and PnIA[A10L] used in this study were tested for the cytotoxicity in RAW264.7 macrophages (Figure 4). PNU282987 at five selected concentrations (12.5 μM, 25 μM, 50 μM, 100 μM, and 200 μM); MLA at five selected concentrations (25 nM, 50 nM, 100 nM, 200 nM, and 400 nM); PnIA[A10L] at five selected concentrations (100 nM, 500 nM, 1 μM, 5 μM, and 10 μM) were tested, and the results are shown in Figure 4a, Figure 4b, and Figure 4c, respectively. The results were analyzed with the one-way ANOVA and Dunn’s multiple comparison method. PNU282987 induced almost no toxicity when the concentration was below 100 μM. When the concentration increased to 200 μM, the cell viability decreased slightly compared with that of the control group, and the cell survival rate was approximately 85%. MLA showed almost no toxicity when the concentration was below 400 nM, and the cell survival rate was greater than 98%. Similarly, the optimal dose of PnIA[A10L] was 10 μM, within this concentration range, the cell viability is greater than 98%. Finally, the safe and reasonable concentrations of PNU282987, MLA, and PnIA[A10L] were identified in our experiment with 100 μM, 400 nM, and 10 μM, respectively.

### 2.5. Regulation of Inflammatory Cytokines in RAW264.7 Cells Mediated through the α7 nAChR Agonist PNU282987

PNU282987, an α7 nAChR agonist, was used to investigate the regulation of α7 nAChR on the inflammatory molecule levels in RAW264.7 macrophages. Changes in TNF-α, IL-6, and IL-1β gene and protein expression levels in inflammatory RAW264.7 cells after activation of the α7 nAChR receptor were detected (Figure 5). In this study, the gene expression levels of the inflammatory factors *Tnf-α*, *Il-6*, and *Il-1β* in macrophages were upregulated 5.3-fold, 202.2-fold, and 293.9-fold, respectively, before and after LPS induction (Figure 5a–c). The expression pattern of these three inflammatory factors also showed a similar trend at the protein level change. The results showed that the macrophage inflammation model was successfully constructed by the introduction of LPS as the stimulus. When macrophages were preincubated with the agonist PNU282987, the gene expression of *Tnf-α* and *Il-1β* was significantly decreased in inflammatory RAW264.7 macrophages, by 32.3% and 51.3%, respectively, but the TNF-α and IL-1β protein levels were not obviously affected (Figure 5a,c). However, for the inflammatory factor IL-6, the expression of both the gene and protein levels were significantly different before and after agonist PNU282987 stimulation. Compared with the LPS-induced inflammation models, the expression levels of the proinflammatory factor IL-6 at the gene and protein levels were significantly reduced, by 59.8% and 65.9%, respectively (Figure 5b). This also means that PNU282987, as a specific agonist of α7 nAChR, indeed downregulated the gene expression levels of *Tnf-α*, *Il-6*, and *Il-1β* in inflammatory cells, but only the protein expression level of IL-6 was significantly downregulated.

### 2.6. Regulation of Inflammatory Cytokines in RAW264.7 Cells via the α7 nAChR Antagonists MLA and α-CTx [A10L]PnIA

MLA and α-CTX [A10L]PnIA were used to further explore the regulatory effect of α7 on the inflammatory response of RAW264.7 macrophages. The results demonstrated that TNF-α, IL-6, and IL-1β gene and protein expression levels changed when MLA and [A10L]PnIA antagonists blocked α7 nAChR in inflammatory RAW264.7 (Figure 6). Compared with the LPS-treated group, MLA and [A10L]PnIA upregulated *Tnf-α* gene expression by 1.5-fold and 1.1-fold, respectively, but there was no significant difference in TNF-α protein levels (Figure 6a). After pretreatment with MLA and [A10L]PnIA, IL-6 gene expression was up-regulated by 29.5% and 12.5% in the LPS-treated group, respectively. MLA and [A10L]PnIA upregulated the IL-6 protein expression level by 17.2% and 6.7%, respectively (Figure 6b). Due to the lack of damage after stimulation, changes in IL-1β protein expression levels could not be detected; only changes in gene expression levels were measured. Stimulated by MLA and [A10L]PnIA, the *Il-1β* gene expression levels in inflammatory macrophages were upregulated by 53.1% and 51.7%, respectively (Figure 6c). In conclusion, the α7 nAChR antagonists MLA and [A10L]PnIA upregulated the gene expression levels of *Tnf-α*, *Il-6,* and *Il-1β* in inflammatory RAW264.7 cells, but only the expression level of the IL-6 protein was increased.

### 2.7. Effects of an Agonist and Antagonists on α7 nAChR Gene Expression in Inflammatory Macrophages RAW264.7

This study also investigated whether the regulation of inflammation by agonist and antagonists is related to the expression of α7 nAChR (Figure 7). qPCR results showed that the α7 nAChR agonist PNU282987 exerted no significant effect on α7 nAChR gene expression levels in normal and inflammatory RAW264.7 cells. Similarly, when the α7 nAChR antagonists MLA and [A10L]PnIA were incubated with normal and the LPS-stimulated inflammatory RAW264.7 cells, there was no statistically significant difference in α7 nAChR mRNA expression levels in the antagonist group and the two group RAW264.7 macrophage. These results indicated that neither the agonist nor antagonists affected the expression of the α7 nAChR gene, and may just modulate the inflammatory response by altering the function of α7 nAChR.

### 2.8. RNA-Seq Analysis of RAW264.7 Inflammatory Regulation Mediated via α7 nAChR Agonists and Antagonists

To further explore the effects of α7 nAChR agonists and antagonists on the expression of inflammation-related genes in inflammatory RAW264.7 cells, RNA-seq analysis was performed on the LPS (200 ng/mL), PNU282987 (100 μM)–LPS (200 ng/mL), MLA (400 nM)–LPS (200 ng/mL), and [A10L]PnIA (10 μM)–LPS (200 ng/mL) groups (Figure 8). We compared the agonist PNU282987–LPS, antagonist MLA–LPS, and [A10L]PnIA–LPS groups with the LPS-only stimulation group (Figure 8a) and found 237 differentially expressed genes in the agonist PNU282987–LPS group compared with the LPS stimulation group, and among these genes, the expression of 135 genes was upregulated, and that of 102 genes was downregulated. There were 103 differentially expressed genes (51 upregulated and 52 downregulated) in the MLA–LPS group compared with the LPS stimulation group and 98 differentially expressed genes (43 upregulated and 55 downregulated) in the [A10L]PnIA–LPS group compared with the LPS group (Figure 8b). Thus, the agonist had a more extensive effect on genes in inflammatory RAW264.7 cells than the antagonists. We further analyzed the differentially expressed genes affected by the agonist PNU282987 and antagonist MAL and [A10L]PnIA using a Venn diagram, and we found that only 14 genes were jointly affected by the agonist and antagonists, indicating that the agonist and antagonists had different effects on the genes in the inflammatory macrophages, and some of these differences were profound. To further analyze the α7 nAChR agonist and antagonist effects on inflammation-related gene pathways, Kyoto Encyclopedia of Genes and Genomes (KEGG) pathway enrichment analysis was performed with the differentially expressed genes in the three treatment groups, and the 10 most enriched pathways were found on the basis of reliability (*q*-value size) (Figure 8c). We found that the three groups of differential genes were indeed enriched in inflammatory pathways, which were all related to the rheumatoid arthritis pathway, and the pathway with differentially expressed genes induced by the antagonistic MLA was closely related to the TNF signaling pathway. We then conducted a heatmap analysis of key inflammation-related genes enriched in inflammatory pathways (Figure 8d) and found that the IL-6 gene was significantly downregulated in the PNU282987–LPS group compared with its expression in the LPS group, was slightly upregulated in the MLA–LPS group, but not significantly upregulated in the [A10L]PnIA treatment group. These results may be because [A10L]PnIA has less α7 nAChR binding activity than MLA. Analysis of the *Il-1β* gene showed that the agonist PNU282987 had no significant effect on *Il-1β*, but the antagonists MLA and [A10L]PnIA enhanced the expression of *Il-1β* in the inflammatory RAW264.7 cells. Interestingly, the expression of the *Il-1α* gene, another inflammatory factor, was also enhanced in the MLA and [A10L]PnIA groups. Although no effect on *Tnf-α* was observed in the three groups in the differential gene analysis, the expression of another tumor necrosis factor Ligand Superfamily member 13 (*Tnfsf13*) was attenuated in the PNU282987–LPS group compared with the LPS group and was increased in the MLA–LPS group. Colony-stimulating factor 2 (*Csf2*), attracted our attention. Our analysis showed that the expression of *Csf2* was enhanced after treatment with PNU282987 in inflammatory RAW264.7 cells, while the expression of the inflammatory RAW264.7 cells treated with antagonist MLA and [A10L]PnIA decreased. In conclusion, the agonist PNU282987 and antagonists MLA and [A10L]PnIA exerted different effects on the RAW264.7 gene in inflammatory macrophages (the agonist has more extensive effects), and the most significant effects were on the inflammatory-related gene *Il-6*. In addition to that of the *Il-6* gene, the expressions of the *Il-α*, *Il-1β*, *Tnfsf13*, and *Csf2* genes were upregulated or downregulated in some groups.

## 3. Discussion

α7 nAChR is widely expressed in many regions of the brain and a variety of nerve cells, and it has become a hotspot in research on many brain diseases, such as Alzheimer’s disease, neuroinflammation, and neuralgia [30]. Stimulated by some inflammatory inducers, astrocytes, microglia, and macrophages in the brain show activation of the NF-κB pathway, which accelerates the maturation and secretion of IL-1β and IL-18, thus recruiting more inflammatory cytokines and cells to promote the development of inflammation and damage in the brain [31,32,33]. Research has shown that, on the one hand, α7 can reduce the release of inflammatory factors by regulating multiple pathways, such as the NF-κB and JAK2/STAT3 pathways, and on the other hand, it can upregulate the glutamate transporter (GLAST) in microglia to enhance glutamate clearance [32,34]. These regulatory mechanisms are key to the neuroprotection of α7 and the treatment of inflammatory diseases in the brain. Studies on α7 nAChR-mediated inflammation outside the central nervous system, research has focused on the modulating mechanisms of the α7 nAChR-mediated cholinergic anti-inflammatory pathway in inflammatory bowel disease, rheumatoid joints, atherosclerosis, and other diseases [35,36]. Recent studies have shown that α7 nAChR exerts an anti-inflammatory effect by inhibiting NF-κB and activating JAK2/STAT3 intracellular signaling pathways to reduce the release of inflammatory factors such as TNF-α, IL-6, and IL-1β [37]; however, Han Q. Q. et al. (2020) found that activation of α7 nAChR can also mediate an increase in interleukin-10 (IL-10) secretion, thus achieving an anti-inflammatory effect [38]. In our study, an inflammatory cell model was constructed with RAW264.7 macrophages induced by LPS in vitro, and the expression level of the α7 nAChR gene on macrophages before and after LPS-induced inflammation was determined by qPCR. All neuronal nAChR subunits were expressed in RAW264.7 cells, and α7 receptor subunits were moderately expressed. After macrophage inflammation, the expression of these subunits showed a slight decrease, but no significant difference was found. qPCR analysis showed that the gene expression of the α7 nAChR receptor gene was not significantly affected by either the α7 nAChR agonist PNU282987 or the antagonists MLA and [A10L]PnIA. These results indicate that the anti-inflammatory effects of the agonist and proinflammatory effects of the antagonists were not realized through the regulation of α7 nAChR receptor gene expression.

IL-6 is a cytokine produced by a variety of immune cells and is functional in a variety of immune cells, such as macrophages, T cells, and B cells. It is a soluble mediator with pleiotropic effects on inflammation, immune response, and hematopoiesis. It can also transmit information, activate and regulate immune cells, promote the expression of interleukin-2 receptor (IL-2R) on the surface of T cells, and enhance the effect of IL-1 and TNF-α on cells, thus playing important roles in the inflammatory response [39,40,41]. IL-6 is involved in the pathological process of various clinical diseases, including bacterial infection, neonatal sepsis, respiratory infections, enteritis, rheumatoid arthritis, and other acute or chronic inflammatory diseases [42,43]. In this study, we found that activation of α7 nAChR with the agonist PNU282987 significantly reduced the gene expression of the inflammatory cytokines *Tnf-α*, *Il-6*, and *Il-1β*, but only IL-6 was significantly decreased at the protein expression level. These results indicated that IL-6 is a vital factor in the anti-inflammatory mechanism through its activation of a7 nAChR. We confirmed this result by using the α7 nAChR-specific antagonist MLA and α-CTx [A10L]PnIA (with an IC_50_ of 12.5 nM against α7 nAChR) [44] and found that the antagonists significantly enhanced the gene and protein expression of IL-6 in inflammatory cells. Therefore, the agonist and antagonists of α7 nAChR regulated inflammation mainly by regulating the expression level of the inflammatory factor IL-6. The specificity that agonists and antagonists only affect IL-6 protein levels may result from post-transcriptional regulation of IL-6. Previous studies have shown that the stability of IL-6 mRNA is regulated by the 3′UTR region. There are many RNA-binding proteins and microRNAs that bind to this region and affect the stability of IL-6 mRNA [45]. Palanisamy et al. (2012) found that TTP, BRF1, BRF2, and other proteins can bind to 3′UTR to accelerate the degradation of IL-6 mRNA [46]. Iwasaki et al. (2011) found that the inhibitor of NF-kB (IkB) kinase (IKK) complex also phosphorylates Regnase-1 and thus destabilizes IL-6 mRNA [47]. Notably, Masuda et al. (2013) found another RNA-binding protein, AT-rich interactive domain-containing protein 5a (Arid5a), which binds to 3′UTR and affects IL-6 expression but does not affect TNF-α expression [48]. These findings suggest that our α7 nAChR agonist PNU282987, antagonist MLA and [A10L]PnIA significantly affect IL-6, which may not only affect its gene transcription but also affect its post-transcriptional stability. However, these speculations need to be tested in future research. Notably, RNA-seq revealed that only 14 genes affected by the α7 nAChR agonist PNU282987 and antagonist MLA and [A10L]PnIA jointly affected inflammatory macrophages, indicating that the agonist and antagonists exert different effects on inflammatory macrophage genes, and in some instances, these differences were very large. Moreover, the agonist PNU282987 affected a wider range of genes.

Interestingly, our RNA-seq analysis showed that *Csf2* gene expression was increased in inflammatory macrophages stimulated by the agonist PNU282987, while stimulation with the antagonists MLA and [A10L]PnIA reduced *Csf2* gene expression levels in inflammatory macrophages. These results may suggest that α7 nAChR is involved in another pathway involved in macrophage inflammation regulation. Macrophages can be classified into the M1 and M2 types, among which the M1 type is a proinflammatory macrophage that can secrete a large number of proinflammatory factors, while the M2 type is an anti-inflammatory macrophage that can release IL-10 and other anti-inflammatory factors to reduce inflammation. An M1 and M2 macrophage can be transformed to acquire the phenotype of the other type, which is an important inflammatory regulatory response that can be leveraged to treat many diseases [49,50]. Studies have shown that the main population of macrophages in many cancers acquire the M2 phenotype [51,52]. Regulating the polarization of M2 macrophages into M1 macrophages can promote the clearance of macrophages on tumor cells, which is a promising combined anticancer therapy strategy. Research has shown that macrophage colony-stimulating factors affect differentiation; for example, colony-stimulating factor 1 (*Csf1*), which regulates the steady state of macrophages, is the main factor involved in monocyte/macrophage lineage differentiation, proliferation, chemokine production, and cell survival [53]. *Csf2* is a glycoprotein that stimulates the proliferation and activation of neutrophil hematopoietic cells, which can induce the proliferation and differentiation of M1 macrophages and enhance antigen presentation [54]. These results suggested that, in this study, α7 nAChR may regulate the polarization of RAW264.7 macrophages by mediating the expression of *Csf2*, thus regulating the level of inflammation. LPS stimulated the M1-type differentiation of RAW264.7 cells, and after α7 nAChR agonist stimulation, the *Csf2* gene expression level of the inflammatory macrophages was increased, which led to the polarization of M1 macrophages into M2 macrophages, leading to the downregulation of inflammatory factors, while the α7 nAChR antagonists had the opposite effect.

## 4. Materials and Methods

### 4.1. Materials

The agonist PNU282987 and antagonist MLA were purchased from Tocris Bioscience (Bristol, UK), and LPS was purchased from Sigma–Aldrich (St. Louis, MO, USA). The [A10L]PnIA linear crude peptide was synthesized by Bankpeptide Biological Technology Co., Ltd. (Hefei, China) and purified in our laboratory.

A CCK-8 kit, DMEM-H medium, Penicillin/Streptomycin Solution, and PBS buffer were purchased from Sangon Biotech (Shanghai, China). FBS was purchased from Procell (Wuhan, China). A FastPure Cell/Tissue Total RNA Isolation Kit, HiScript II Q Select RT SuperMix for qPCR (+gDNA wiper), 2 × Taq Plus Master Mix Ⅱ, and ChamQ Universal SYBR qPCR Master Mix were obtained from Vazyme (Nanjing, China). A TNF-α ELISA kit, IL-6 ELISA kit, and IL-1β ELISA kit were purchased from Thermo Scientific (Waltham, MA, USA).

### 4.2. Cell Culture

RAW264.7 mouse mononuclear macrophage cells were purchased from the BNCC Cell Bank (Beijing, China). RAW264.7 cells were cultured in a cell culture flask or dish with 10% (*v*/*v*) inactivated bovine serum, 100 units/mL penicillin, 100 μg/mL streptomycin, 4 mM l-glutamine and phenol red DMEM high glucose medium in 5% CO_2_, 37 °C cell incubator. For normal round or oval cells, 1:4 cell passage was carried out by blowing. In general, the same batch of cells will be used within 10 generations (about half a month), in case of an abnormal cell state, such as many pseudopodia, a new cell is resurrected to ensure the stability of the experiment.

### 4.3. Identification of nAChR Subunits on RAW264.7 Macrophages

Four groups of biologically replicated RAW264.7 mononuclear macrophages were cultured. When the cells reached a density of 2–5 × 10^5^ cells, the total RNA of four groups of RAW264.7 cells was extracted with a rapid extraction cell total RNA kit (Vazyme, China), and the concentration was determined with a Nanodrop 2000 spectrophotometer (Thermo Scientific, Waltham, MA, USA). The total extracted cellular RNA was used as a template for in vitro transcription. The procedure was performed with a reverse transcription kit (Vazyme, Nanjing, China) following the manufacturer’s instructions. The primers of different nAChR subunits were designed with Snap Gene software 3.0 (San Diego, CA, USA) and synthesized by Sangon Biotech Co., Ltd. (Shanghai, China). All primers are shown in Appendix A. In a 50 μL PCR system, 5 μL of cDNA was used as a template, and 25 μL of 2 × Taq Plus Master Mix Ⅱ was added (Vazyme, Nanjing, China). PCR amplification was performed with a cycling protocol composed of an initial denaturation step at 95 °C for 3 min, 35 cycles at 95 °C for 15 s, 55 °C for 40 s, and 72 °C for 50 s, and termination with a final extension at 72 °C for 10 min.

### 4.4. Synthesis of the α7 nAChR Specific Antagonist [A10L]PnIA

The α-CTx [A10L]PnIA (GCCSLPPCALNNPDYC^#^, ^#^ represents an amidated C-terminus) linear crude peptide was synthesized by Bankpeptide Biological Technology Co., Ltd. (Hefei, China). As shown in Figure 1, the Cystine (Cys) residues were protected in pairs with *S*-acetamidomethyl (Acm) on Cys2 and Cys4 (the second and fourth Cys residue). The crude peptide was purified by RP-HPLC with a Waters 2695 preparative device (Waters, Milford, MA, USA). The purified linear [A10L]PnIA was oxidized in two steps as described previously [55]. In brief, the first disulfide bridge was formed in 20 mM K_3_[Fe(CN)_6_] and 0.1 M Tris-HCl buffer (pH 7.5). After the reaction, the monocyclic peptide products were purified by preparative chromatography. The purification condition was a linear-gradient elution with 5–50% Buffer B (0.1% TFA/99.9% ACN) for 50 min. Buffer A solution was 0.1% TFA in ddH_2_O. The flow rate was 10 mL/min, and the wavelength of the UV monitor was set to 214 nm. The second disulfide bridge was formed in the I_2_ solution. The final oxidized folding product was purified by RP-HPLC with a C18 column. The final active [A10L]PnIA was purified by RP-HPLC with an analytical C18 column (Vydac, Waters, 4.6 mm × 250 mm, 5 μm) using a linear gradient of 10–55% Buffer B in 35 min, and the flow rate was 0.8 mL/min. The molecular mass of [A10L]PnIA was identified by ESI-MS (Acquity H Class-Xevo TOD, Waters, Milford, MA, USA).

### 4.5. CCK-8 Cytotoxicity Assay

The inflammatory inducer LPS, α7 nAChR agonist (PNU282987), and antagonist (MLA, [A10L]PnIA) used in this study were all evaluated for cytotoxicity by CCK-8 assay to ensure the safety and effectiveness of the concentration used in the experiments. RAW 264.7 cells were cultured, subcultured, and seeded into 96-well plates at a density of 3000 cells/well. After 24 h of culture, LPS, PNU282987, MLA, and PnIA[A10L] were administered at five different concentrations and incubated for 24 h. Five replicates were set for each concentration. After the original medium was discarded, 100 μL of culture solution (CCK-8: DMEM-H = 1:10) containing CCK-8 reagent was added to each well and incubated at 37 °C for 1–2 h. Then, the absorbance value was detected at a 450 nm wavelength by using a SpectraMax M2 microplate reader (Molecular Devices, Sunnyvale, CA, USA) to calculate the effect of the drug on cell viability. The viability of the cells was calculated by the equation-relative cell viability (%) = (OD of treatment group − OD of the control group)/(OD of the blank group − OD of the control group) using GraphPad Prism 8 software (San Diego, CA, USA).

### 4.6. qPCR and ELISAs

RAW 264.7 cells were subcultured and seeded in 6-well plates at a cell density of 2–5 × 10^4^. After culturing for 24 h and serum-free treatment for 2 h, the cells were treated with α7 nAChR agonist or antagonist for 6–8 h and cultured with 200 ng/mL LPS stimulation for 12–18 h. Total RNA was extracted from different groups of cells with a rapid extraction cell total RNA kit (Vazyme, Nanjing, China), and the concentration was determined with a Nanodrop 2000 spectrophotometer (Thermo Scientific, Waltham, MA, USA). The total extracted cellular RNA was used as a template for in vitro transcription. The procedure was performed with a reverse transcription kit (Vazyme, Nanjing, China) following the manufacturer’s instructions. *Actin* and *Gapdh* were used as reference genes to conduct relative qPCR for nAChRs subunit genes. The primers for different nAChR subunits were designed with Snap Gene software 3.0 (San Diego, CA, USA) and synthesized by Sangon Biotech Co., Ltd. (Shanghai, China). The primers for real-time quantitative PCR are shown in Appendix A. In a 20 μL qPCR system, 2 μL of cDNA was used as a template, and 10 μL ChamQ Universal SYBR qPCR Master Mix (Vazyme, Nanjing, China) was added. PCR amplification was performed with a cycling protocol composed of an initial denaturation step at 95 °C for 5 min, 45 cycles at 95 °C for 10 s, 55 °C for 20 s, and 72 °C for 30 s, and termination with a final extension at 65 °C for 10 min.

The protein expression levels of the inflammatory cytokines TNF-α, IL-6, and IL-1β were measured using commercial ELISA kits (Thermo, Waltham, MA, USA). RAW264.7 cells were cultured, subcultured, and inoculated in 24-well plates with 2–5 × 10^4^ cells per well. The cells were stimulated with an agonist and antagonists of α7 nAChR for 6–8 h and then stimulated with 200 ng/mL LPS for 12–18 h. The cell culture medium of the different groups on each 24-well plate was collected and centrifuged at 12,000 rpm/min at 4 °C for 10 min, and the supernatant was retained. According to the instructions of each ELISA kit, the absorbance value at 450 nm with a SpectraMax M2 microplate reader (Molecular Devices, Sunnyvale, CA, USA), and the content of the corresponding inflammatory factors was calculated according to the standard curve. The detection ranges of TNF-α, IL-6 and IL-1β were 7.8–500 pg/mL, 7.8–500 pg/mL, and 7.8–500 pg/mL, respectively.

### 4.7. RNA-Seq Analysis

Total RNA was isolated and extracted from RAW264.7 macrophages in different treatment groups for the preparation of a cDNA library, and subsequent on-machine detection was carried out by Biomarker Technologies (Beijing, China). Analysis of differential gene expression between groups was performed with the edgeR package in R software with the FDR criterion set to 5% and the fold change threshold set to a value greater than 1.5. The BMKCloud biological cloud computing platform was used for further analysis and mapping, and the topGO R language software package was used to further analyze the KEGG pathways enriched with differentially expressed genes.

### 4.8. Statistical Analysis

Data analysis was performed using GraphPad Prism 8 software (San Diego, CA, USA) for one-way analysis of variance, and Dunnett’s multiple comparisons test was used for comparisons between groups. Statistical significance is indicated by * and #. * and # indicate a *p*-value < 0.05, ** and ## indicate a *p*-value < 0.01, *** and ### indicate a *p*-value < 0.001, **** and #### indicate a *p*-value < 0.0001; the error bars in all figures indicate the means ± SEM (*n* = 3–6).

## 5. Conclusions

This study showed that the differentially expressed genes induced by the α7 nAChR agonist PNU282987 or antagonist MLA or α-CTx [A10L]PnIA were enriched in inflammatory pathways, but the range of genes affected by the agonist and antagonists was very different. IL-6 is a key inflammatory factor in the regulation of α7 nAChR-mediated macrophage inflammation. Changes in IL-6 expression relative to that of TNF-α and IL-1β were significant at both the gene and protein levels, and this effect was not related to the expression level of the α7 nAChR itself. Another interesting finding was that both the α7 nAChR agonist and antagonists exerted significant effects on *Csf2* expression, which may suggest a new mechanism of inflammation mediated by α7 nAChR. This study provides a pharmacological basis for the future discovery of agonists and antagonists of α7 nAChR and their application in the study of the α7 nAChR-mediated inflammatory response.

## Figures and Tables

**Figure 1 marinedrugs-20-00200-f001:**
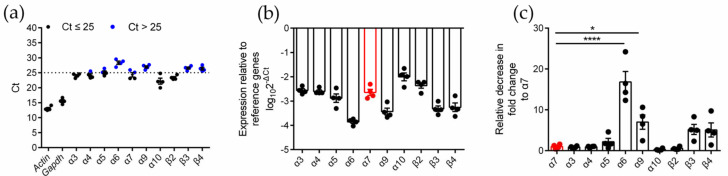
qPCR for various nAChR subunits gene expression analysis on normal RAW264.7 macrophages. The experimental procedure was described in Methods. (**a**): The Ct value of mRNA measured in four replicate normal cell groups. Each data point represents the average of three technical replicates of each gene from a single group. Blue circles in (**a**) indicate Ct values > 25, whereas black circles indicate values ≤ 25. (**b**): The expression levels of nAChR subunit mRNAs relative to those of the reference genes *Gapdh*. (**c**): The decrease in fold change of other nAChR subunits relative to α7 subunits. ns, not significant or *p* > 0.05; * *p* ≤ 0.05; **** *p* ≤ 0.0001; compared to the *Chrna-7* 2^ΔCt^; data indicate mean ± SEM; the values and statistical comparisons are shown in Table 1.

**Figure 2 marinedrugs-20-00200-f002:**
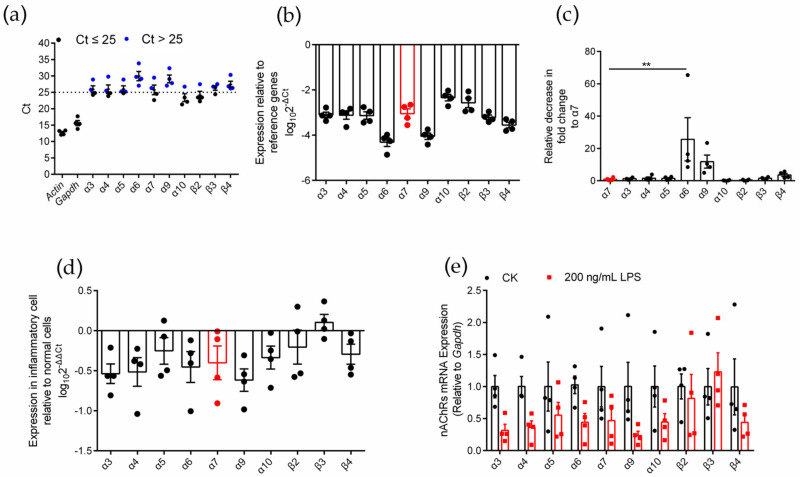
qPCR for various nAChR subunits gene expression analysis on LPS induced inflammatory RAW264.7 macrophages. (**a**): The Ct value of mRNA measured in four replicate normal cell groups. Each data point represents the average of three technical replicates of each gene from a single group. Blue circles in panel (**a**) indicate Ct values > 25 whereas black circles indicate values ≤ 25. (**b**): The expression levels of nAChR subunit mRNAs relative to those of the reference genes *Gapdh*. (**c**): The decrease in fold change of other nAChR subunits relative to α7 subunits. (**d**): Expression of each nAChR subunit gene in inflammatory cells relative to normal cells. Positive values indicate higher levels of expression in the inflammatory cell and negative values indicate higher levels in the normal cell for each gene. (**e**): A pairwise comparison for each nAChR subunit gene of the relative expression levels present in normal and inflammatory cells. ns, not significant or *p* > 0.05; ** *p* ≤ 0.01; the error bars in all figures indicate SEM; the values and statistical comparisons are shown in Table 1.

**Figure 3 marinedrugs-20-00200-f003:**
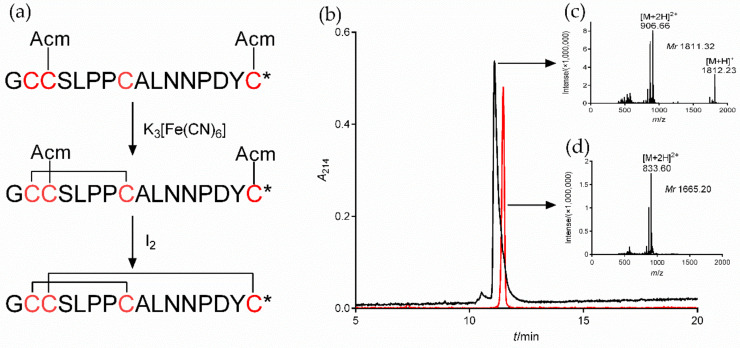
Oxidative folding, HPLC, and ESI-MS of α-CTx [A10L]PnIA. (**a**): Oxidative folding process of the linear crude [A10L]PnIA. The first step is oxidative folding in K_3_[Fe(CN)_6_] buffer, and the second step is oxidative folding in I_2_ solution under the protection of nitrogen. * amidated COOH terminal. (**b**): The HPLC profile of linear and Acm-protected (black) and oxidized (red) [A10L]PnIA. The linear and oxidized [A10L]PnIA were analyzed by reversed-phase analytical HPLC on a Vydac C18 column using a linear gradient of 10% to 55% Buffer B (0.1% TFA and 99.9% acetonitrile) over 30 min. Absorbance was monitored at 214 nm. (**c**): ESI-MS data of linear and Acm-protected [A10L]PnIA, with a monoisotopic mass of 1811.32 Da. (**d**): ESI-MS data of [A10L]PnIA, with a monoisotopic mass of 1665.20 Da.

**Figure 4 marinedrugs-20-00200-f004:**
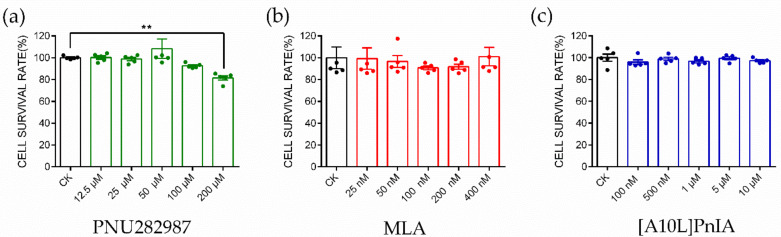
CCK8 experiment to identify the cytotoxicity of α7 agonist and antagonist. CCK8 experiment to evaluate the cytotoxicity of PNU282987 (**a**), MLA (**b**), and [A10L]PnIA (**c**) at five different concentration conditions. Each data represents mean ± SEM (*n* = 5). Significance was determined using a one-way ANOVA with a Dunnett’s multiple comparisons test; ns, not significant or *p* > 0.05; ** *p* ≤ 0.01 vs Control (CK).

**Figure 5 marinedrugs-20-00200-f005:**
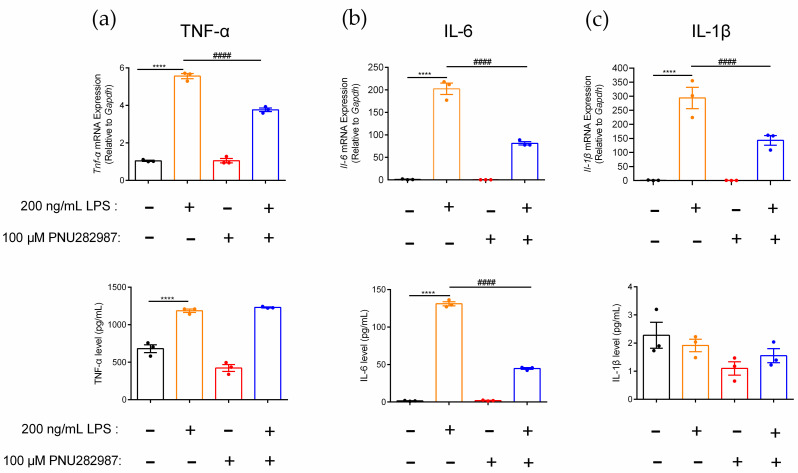
Effects of PNU282987 on inhibition of proinflammatory cytokine expression in inflammatory RAW264.7 cells. After the cells were treated with PNU282987 (100 μM) for 6–8 h, with or without LPS (200 ng/mL) over a further 12–16 h. (**a**–**c**): The effects of PNU282987 on mRNA gene and protein expression level of proinflammatory cytokine TNF-α, IL-6, and IL-1β of inflammatory RAW264.7, respectively. qPCR was used to determine the mRNA gene expression of *Tnf-α*, *Il-6*, and *Il-1β* in the cells, and the protein expression of TNF-α, IL-6, and IL-1β in the culture medium was detected with an ELISA kit. ns, not significant or *p* > 0.05; ****, #### *p* ≤ 0.0001, compared to the control and LPS-treated groups respectively, by one-way ANOVA followed by the Dunnett’s multiple comparisons tests, the error bars in all figures indicate SEM (*n* = 3).

**Figure 6 marinedrugs-20-00200-f006:**
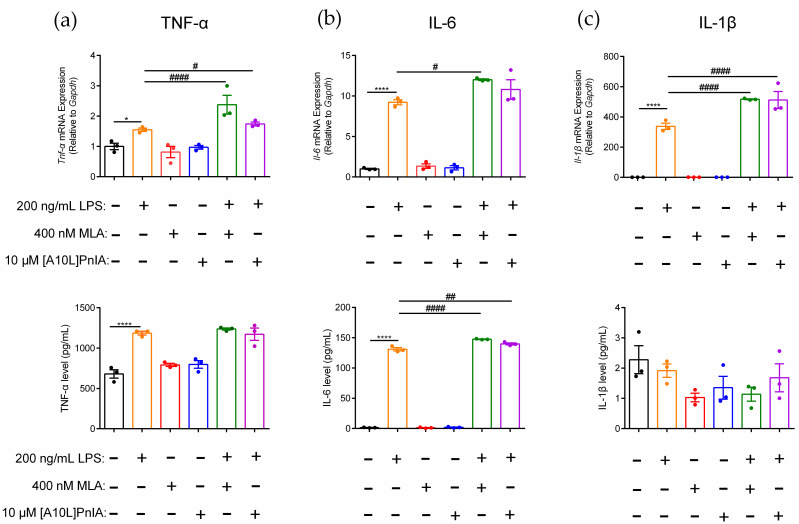
Effects of antagonists MLA and [A10L]PnIA on the promotion of proinflammatory cytokine expression in inflammatory RAW264.7 cells. After the cells were treated with MLA (400 nM) and [A10L]PnIA (10 μM) for 6–8 h, with or without LPS (200 ng/mL) over a further 12–16 h. (**a**–**c**): The effects of MLA and [A10L]PnIA on mRNA gene and protein expression level of proinflammatory cytokine TNF-α, IL-6, and IL-1β of inflammatory RAW264.7, respectively. qPCR was used to determine the mRNA gene expression of *Tnf-α*, *Il-6*, and *Il-1β* in the cells, and the protein expression of TNF-α, IL-6, and IL-1β in the culture medium was detected with an ELISA kit. ns, not significant or *p* > 0.05; *, # *p* ≤ 0.05; ## *p* ≤ 0.01; ****, #### *p* ≤ 0.0001, compared to the control and LPS-treated groups respectively, by one-way ANOVA followed by the Dunnett’s multiple comparisons tests, the error bars in all figures indicate SEM (*n* = 3).

**Figure 7 marinedrugs-20-00200-f007:**
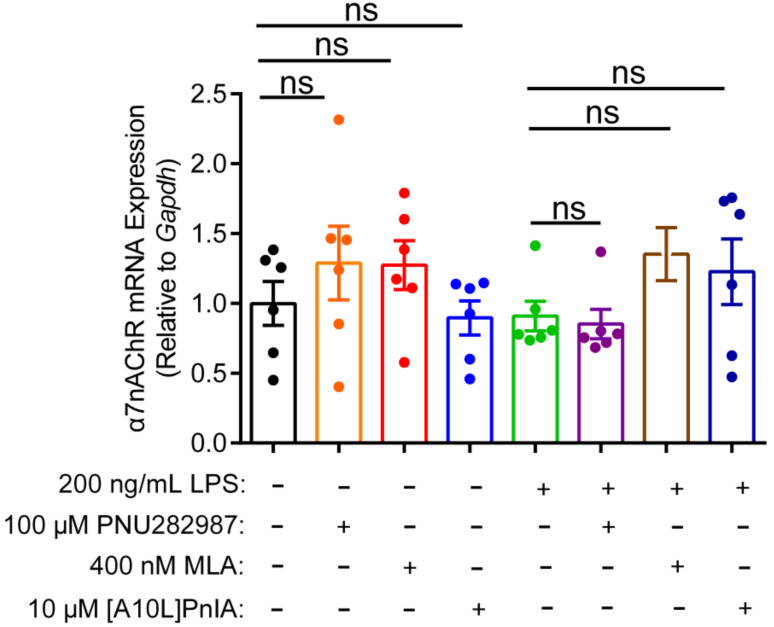
The effect of agonists and antagonists on α7 nAChR mRNA expression in RAW264.7 macrophages. After the cells were treated with α7 nAChR agonist [PNU282987 (100 μM)] and antagonists [MLA (400 nM) and [A10L]PnIA] for 6–8 h, with or without LPS (200 ng/mL) over further 12–16 h. The effects of LPS (200 ng/mL), PNU282987 (100 μM), MLA(400 nM), [A10L]PnIA (10 μM) on α7 nAChR mRNA expression were identified by using qPCR, ns, not significant or *p* > 0.05, compared to the control and LPS-treated groups respectively, by one-way ANOVA followed by the Dunnett’s multiple comparisons test. All data represent mean ± SEM (*n* = 6).

**Figure 8 marinedrugs-20-00200-f008:**
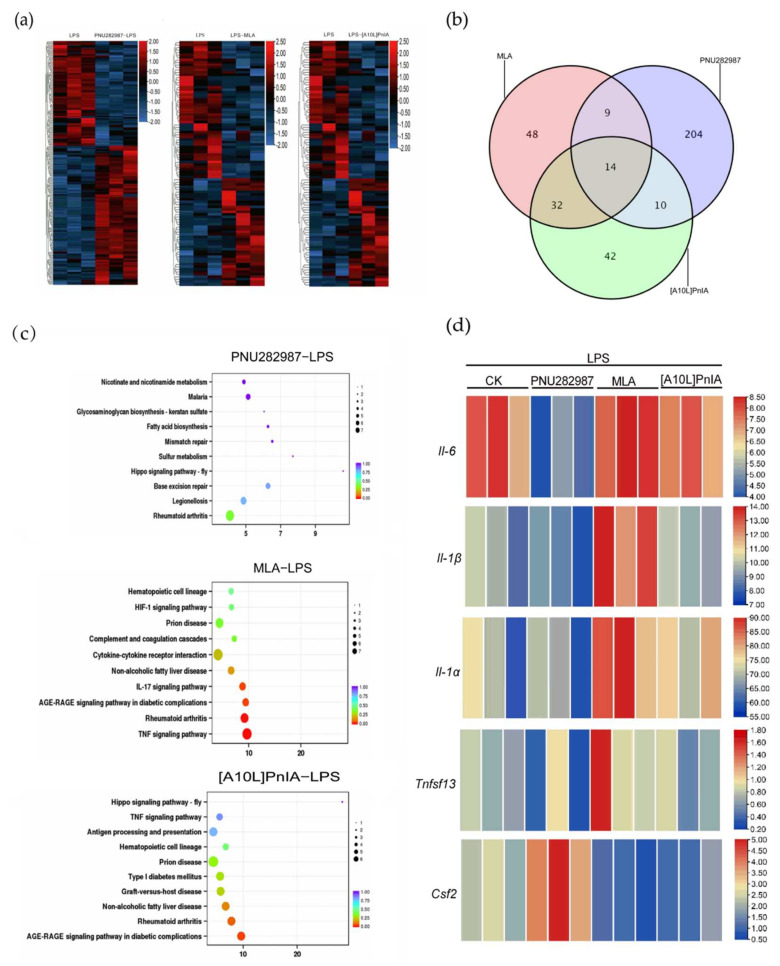
α7 nAChR agonist PNU282987 and antagonists MLA and [A10L]PnIA regulate inflammatory RAW264.7 cell gene expression by RNA sequencing. Differential gene analysis was performed between PNU282987 (100 μM) − LPS (200 ng/mL), MLA (400 nM) − LPS (200 ng/mL), [A10L]PnIA (10 μM) − LPS (200 ng/mL) and LPS (200 ng/mL) groups, respectively. (**a**): The heat map of genes with adjusted *p*-value < 0.05, Analysis by DESeq R package. (**b**): Venn diagram showing the overlap between PNU282987-induced inflammation RAW264.7 cells differential genes and MLA and [A10L]PnIA-induced inflammation RAW264.7 cells differential genes. (**c**): KEGG pathway analysis was performed between PNU282987 (100 μM) − LPS (200 ng/mL), MLA (400 nM) − LPS (200 ng/mL), [A10L]PnIA (10 μM) − LPS (200 ng/mL) and LPS (200 ng/mL) groups, respectively, Analysis by topGO R package, The abscissa is Enrichment Factor; *q*-value is *p*-value after correction of multiple hypothesis testing; the size of the circle is the count of genes enriched in the pathway. (**d**): Heat map analysis of five key genes.

**Table 1 marinedrugs-20-00200-t001:** Cells from four replicate groups were analyzed individually by qPCR.

Normal Cell					Inflammatory Cell				Inflammatory Cell Relative to Normal Cell	
Gene	Ct	*n*	log_10_2^−(ΔCt)^	Relative decrease in fold change to α7	Ct	*n*	log_10_2^-(ΔCt)^	Relative decrease in fold change to α7	log_10_2^−(ΔΔCt)^	Fold-difference
*Actin*	13.0 ± 0.7	4	nd	nd	12.7 ± 0.3	4	nd	nd	nd	nd
*Gapdh*	15.5 ± 0.9	4	nd	nd	15.6 ± 0.8	4	nd	nd	nd	nd
*Chrna-3*	24.1 ± 0.3	4	−2.6 ± 0.1	0.9 ± 0.1 ^ns^	25.9 ± 1.0	4	−3.1 ± 0.1	1.3 ± 0.3 ^ns^	−0.5 ± 0.1	−3.9 ± 1.0 ^ns^
*Chrna-4*	24.2 ± 0.5	4	−2.6 ± 0.1	0.9 ± 0.1 ^ns^	26.0 ± 1.3	4	−3.1 ± 0.2	1.6 ± 0.8 ^ns^	−0.5 ± 0.2	−4.4 ± 2.2 ^ns^
*Chrna-5*	25.1 ± 0.5	4	−2.9 ± 0.2	2.2 ± 0.8 ^ns^	26.0 ± 1.0	4	−3.1 ± 0.2	1.5 ± 0.5 ^ns^	−0.3 ± 0.2	−1.7 ± 1.1 ^ns^
*Chrna-6*	28.3 ± 0.6	4	−3.9 ± 0.1	16.8 ± 2.6 ****	29.9 ± 1.4	4	−4.3 ± 0.2	25.7 ± 13.3 **	−0.5 ± 0.2	−3.0 ± 2.1 ^ns^
*Chrna-7*	24.3 ± 0.7	4	−2.6 ± 0.1	1.0 ± 0.2	25.8 ± 1.5	4	−3.0 ± 0.2	1.0 ± 0.5	−0.4 ± 0.2	−3.5 ± 1.6 ^ns^
*Chrna-9*	26.9 ± 0.4	4	−3.4 ± 0.1	7.0 ± 1.7 *	29.1 ± 1.2	4	−4.0 ± 0.1	11.8 ± 4.0 ^ns^	−0.6 ± 0.1	−4.7 ± 1.7 ^ns^
*Chrna-10*	22.2 ± 1.0	4	−2.0 ± 0.2	0.3 ± 0.1 ^ns^	23.4 ± 1.2	4	−2.3 ± 0.1	0.2 ± 0.1 ^ns^	−0.3 ± 0.1	−2.6 ± 0.9 ^ns^
*Chrnb-2*	23.4 ± 0.8	4	−2.4 ± 0.1	0.6 ± 0.2 ^ns^	24.2 ± 1.1	4	−2.6 ± 0.2	0.5 ± 0.2 ^ns^	−0.2 ± 0.2	−1.6 ± 1.3 ^ns^
*Chrnb-3*	26.5 ± 0.4	4	−3.3 ± 0.1	5.2 ± 1.2 ^ns^	26.3 ± 0.7	4	−3.2 ± 0.1	1.6 ± 0.3 ^ns^	0.1 ± 0.1	0.9 ± 0.8 ^ns^
*Chrnb-4*	26.4 ± 0.4	4	−3.3 ± 0.2	5.1 ± 1.7 ^ns^	27.4 ± 1.0	4	−3.6 ± 0.1	3.6 ± 1.0 ^ns^	−0.3 ± 0.1	−2.2 ± 0.6 ^ns^

Positive values for comparisons of gene expression between a normal cell and inflammation cell indicate greater relative abundance in the normal cell. ‘*n*’ value indicates the number of cell samples tested. Ct, cycle threshold; nd, not determined. Significance was determined using a one-way ANOVA with a Dunnett’s multiple comparisons tests; ns, not significant or *p* > 0.05; * *p* ≤ 0.05; ** *p* ≤ 0.01; **** *p* ≤ 0.0001; compared to the *Chrna-7* 2^ΔCt^ (relative decrease in fold change to α7) or pairwise comparison for each nAChR subunit gene of the relative expression levels present in normal and inflammatory cells (fold-difference); data indicate mean ± SEM.

## Data Availability

The data presented in this study are available on request from the corresponding author. The data are not publicly available due to need for further research.

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
