# Peer review of "Inflammation Regulation via an Agonist and Antagonists of α7 Nicotinic Acetylcholine Receptors in RAW264.7 Macrophages"

_marinedrugs, 2022, doi:10.3390/md20030200_

Round 1

Reviewer 1 Report

nAChRs, in particular the a7 subtype is known to be expressed in macrophages and are implicated in anti-inflammatory pathway. While a7 is extensively studied for its ion channel properties and its role in the central nervous system, its role in the peripheral tissues, including macrophages, is understudied and the field is relatively new. Here authors indicate the potential role of a7 nAChR in modulating macrophage switch from pro-inflamatory to anti-inflamatory cells. Authors use qRT-PCR to establish the presence of nAChR subtypes, including the a7 in RAW264.7 macrophage cell line. The authors quantified nAChR subtypes in inflammatory RAW264.7 cells, determine the effect of a7 nAChR activation, using subtype selective ligands, on proinflammatory cytokines at mRNA level and at protein level, and performed transcriptome analysis to understand the pathways involved in the phenotype. The study is significant considering the importance to determine the role of a7 nAChRs subtype in modulating cholinergic inflammatory pathway and has broader implications in the field. Here are some of the reviews to improve the work.

Reviews:

  • 1: mRNA expression of nAChR subtypes was either relative to house keeping gene or to the a7. Did authors consider quantifying absolute mRNA levels of the subtypes using a standard curve?
  • In figure 3 what was the rational behind the concentrations used in the study? can authors discuss EC50 for the agonist and IC50 of the antagonists? There is reference only once in the discussion for α-CTx [A10L]PnIA, but can be extended to other ligands.
  • Figure 5: Reduction in mRNA levels was observed for all the proinflammatory cytokines tested in the study, with the use of PNU282987, but protein levels of only IL-6 were significantly lowered. While, authors acknowledge the significance of IL-6, can authors discuss the significance of this specificity?
  • While further increase in proinflammatory, at least at an mRNA level, with use of antagonists is interesting, what is surprising is that the experiments involving antagonists are in the absence of any agonist. What is the significance of use of antagonists if the receptors are not activated prior to the application of antagonist? To establish the specificity, it is imperative to include experiments were agonist is used along with an antagonist and LPS to include inflammation.

A general comment is to improve the sentence construction, avoid repetition, mainly while explaining the results. At times it appears that results are discussed while the focus on significance is not evident. Some minor changes, but not limited to.

Line 151: change from Figure 5 to Figure 3.

Line 178: change ‘fold-less’ to relative decrease in fold change.

Line 179: change from ‘tended to be’ to 'tend to'.

Reviewer 2 Report

Tan and colleagues have presented an original paper entitled "Inflammation Regulation via an Agonist and Antagonists of α7 nAChRs in RAW264.7 Macrophages".

The authors present a fairly interesting work that considers a potential pharmacological approach using agonists and antagonists of the a7nAChR suggesting new mechanisms of inflammation mediated by the nicotinic receptor.

The experimental approaches and the results presented are pretty well discussed, but some clarifications are needed to make the paper publishable, so some changes are suggested.

  1. The authors use a stabilized macrophage line such as RAW264.7 for their studies. Materials and methods sections are missing exactly how the cells are maintained, the media used, and above all, at what passages these cultures are. It is known that all cell lines, RAW264.7 included, lose their peculiar characteristics if they are kept for several passages, so if they are kept for a long time, they can negatively influence the reliability of the data. For this reason, the initial question is whether these characteristics have been guaranteed precisely to give credit to the data presented in the present paper. Have you taken these variables into account for your experiments?
  2. Being macrophage cells, is it possible to show images of the cells and if any morphological changes are observed in the case of the various treatments?
  3. The graphs in Figures 1 and 4 presented under table 1 appear somewhat redundant. However, in figure 1 (a), what has been indicated in the abscissa is not clear. Please specify.
  4. Furthermore, it would be advisable to insert the same figure legend that the columns corresponding to alpha7 are highlighted in red. Also, towards what were the p values in the legend of tab 1 calculated?
  5. In general, for easier reading, it would be advisable to insert which value of p corresponds to the various * or # in every single legend (figures and tables). ** or ##, *** or ###, **** or #### (among other things not mentioned in the statistical section of materials and methods).
  6. The agonists and antagonists presented in fig 2 and 3 will be used for the experiments described in figure 5, after indicating the differential gene expression analysis of various nAChR subunits in inflammatory RAW, as explained in paragraphs 2.4 and Fig 4. Probably for do not to create confusion and sudden changes in the sequence of events that are being described, it is advisable to be able to move figs 2 and 3 before 5 and 4 before 2. In this way, 4 becomes 2, and 2 and 3 become 3 and 4 respectively, keeping the 5 in place. This would also involve moving sections 2.2 and 2.3 after 2.4. In this way, after table 1, there will be graphs of the cells' untreated cells (fig 1) after stimulation with LPS (fig 2) and then the treatments with agonist or a7 nAChR receptor antagonists.
  7. In Figures 5 and 6, the data labels are inserted at the top of the graphs. To make reading more manageable and more immediate in this case, I would recommend placing the corresponding treatments on the abscissa.
  8. In section 2.7 and Figure 7, the absence of effects on inflammation by agonists or antagonists of nicotinic receptors is considered. While not the predominant argument, perhaps it might be helpful to include a slight hint of the LPS effect on the regulation of the a7 nAChR. It would also be advisable to build graphs by combining control and treatment (LPS, agonists, antagonists). This could help the reader without inferring the results by comparing the ordinate axes of different graphs.
  9. Have you, by any chance, considered confirming these data on a highly purified microglia culture?

Minor revision

Why is section 2.2 the text in bold?

Round 2

Reviewer 2 Report

I thank the authors who modified the article, following the reviewers' advice, changing the text, figures and legends. Now the paper is ready to be published.

Author Response

Thank you for your comments.